# Italian Survey on Endoscopic Biliary Drainage Approach in Patients with Surgically Altered Anatomy

**DOI:** 10.3390/medicina60030472

**Published:** 2024-03-13

**Authors:** Aurelio Mauro, Cecilia Binda, Alessandro Fugazza, Giuseppe Vanella, Vincenzo Giorgio Mirante, Stefano Mazza, Davide Scalvini, Ilaria Tarantino, Carlo Fabbri, Andrea Anderloni

**Affiliations:** 1Gastroenterology and Digestive Endoscopy Unit, Fondazione IRCCS Policlinico San Matteo, Viale Camillo Golgi 19, 27100 Pavia, Italy; s.mazza@smatteo.pv.it (S.M.); davide.scalvini01@universitadipavia.it (D.S.); a.anderloni@smatteo.pv.it (A.A.); 2Gastroenterology and Digestive Endoscopy Unit, Forlì-Cesena Hospitals, AUSL Romagna, 47121 Forlì, Italy; cecilia.binda@auslromagna.it (C.B.); carlo.fabbri@auslromagna.it (C.F.); 3Division of Gastroenterology and Digestive Endoscopy, Humanitas Research Hospital-IRCCS, Via Manzoni 56, Rozzano, 20089 Milan, Italy; alessandro.fugazza@humanitas.it; 4Pancreatobiliary Endoscopy and Endosonography Division, Pancreas Translational & Clinical Research Center, IRCCS San Raffaele Scientific Institute, 20132 Milan, Italy; vanella.giuseppe@hsr.it; 5IRCCS San Raffaele Scientific Institute, Pancreatobiliary Endoscopy and Endosonography Division, 20132 Milan, Italy; 6Gastroenterology and Digestive Endoscopy Unit, Azienda USL—IRCCS di Reggio Emilia, 42122 Reggio Emilia, Italy; vincenzogiorgio.mirante@ausl.re.it; 7Department of Internal Medicine, University of Pavia, 27100 Pavia, Italy; 8Endoscopy Service, Department of Diagnostic and Therapeutic Services, IRCCS-ISMETT, 90127 Palermo, Italy; itarantino@ismett.edu

**Keywords:** altered anatomy, biliary drainage, Roux-en-Y, Billroth-II, ERCP, EUS-guided biliary drainage, device-assisted ERCP

## Abstract

*Background and Objectives*: Biliary drainage (BD) in patients with surgically altered anatomy (SAA) could be obtained endoscopically with different techniques or with a percutaneous approach. Every endoscopic technique could be challenging and not clearly superior over another. The aim of this survey is to explore which is the standard BD approach in patients with SAA. *Materials and Methods*: A 34-question online survey was sent to different Italian tertiary and non-tertiary endoscopic centers performing interventional biliopancreatic endoscopy. The core of the survey was focused on the first-line and alternative BD approaches to SAA patients with benign or malignant obstruction. *Results*: Out of 70 centers, 39 answered the survey (response rate: 56%). Only 48.7% of them declared themselves to be reference centers for endoscopic BD in SAA. The total number of procedures performed per year is usually low, especially in non-tertiary centers; however, they have a low tendency to refer to more experienced centers. In the case of Billroth-II reconstruction, the majority of centers declared that they use a duodenoscope or forward-viewing scope in both benign and malignant diseases as a first approach. However, in the case of failure, the BD approach becomes extremely heterogeneous among centers without any technique prevailing over the others. Interestingly, in the case of Roux-en-Y, a significant proportion of centers declared that they choose the percutaneous approach in both benign (35.1%) and malignant obstruction (32.4%) as a first option. In the case of a previous failed attempt at BD in Roux-en-Y, the subsequent most used approach is the EUS-guided intervention in both benign and malignant indications. *Conclusions*: This survey shows that the endoscopic BD approach is extremely heterogeneous, especially in patients with Roux-en-Y reconstruction or after ERCP failure in Billroth-II reconstruction. Percutaneous BD is still taken into account by a significant proportion of centers in the case of Roux-en-Y anatomy. The total number of endoscopic BD procedures performed in non-tertiary centers is usually low, but this result does not correspond to an adequate rate of referral to more experienced centers.

## 1. Introduction

The number of requests for biliary drainage (BD) in patients with a surgically altered anatomy (SAA) (e.g., Billroth-II or Roux-en-Y reconstruction) are increasing at endoscopic departments considering the epidemiological trend of bariatric surgery and improved survival of patients with operated upper-GI cancers [1,2]. Biliary obstruction may be a consequence of both benign and malignant conditions. SAA is defined when there is no continuity between the stomach, or its remnant, and the duodenum. Surgical reconstructions that lead to this type of SAA are Billroth-II, Roux-en-Y and pancreaticoduodenectomy [1].

Endoscopic BD in normal anatomy is usually performed with endoscopic retrograde cholangiopancreatography (ERCP) and, as an alternative, more recently, with endoscopic ultrasound (EUS)-guided BD. ERCP with the standard duodenoscope is not feasible in Roux-en-Y anatomies considering the length of the afferent limb and is very challenging for the other type of SAA. When standard ERCP is not feasible in patients with SAA, there are different available options for achieving endoscopic BD. There are some luminal techniques performed with forward-viewing scopes (i.e., colonoscope or operative gastroscope) [3,4] or with device-assisted enteroscopy (DAE) ERCP performed with different types of enteroscope (double-balloon, single-balloon or spiral enteroscope) [5]. More recently, EUS-guided BD has been also proposed for patients with SAA [6,7]. Different types of interventional EUS procedures could be performed in patients with SAA. Direct EUS-guided BD could be attempted in Billroth-II reconstruction if the afferent loop is not angulated and allows the passage of the echoendoscope up to the duodenal stump in order to visualize the target for the biliary drainage. Antegrade EUS intervention is performed from the gastric remnant or the jejunal loop to the left liver lobe if the intrahepatic ducts are sufficiently dilated. In this situation, it is possible to directly drain the biliary tree with a hepaticogastrostomy or with a transpapillary or transanastomotic stent. Finally, interventional EUS allows the possibility to create a rapid communication between the gastric remnant and the afferent loop in Roux-en-Y reconstruction with the execution of a gastrojejunal anastomosis with LAMS placement and a subsequent execution of the ERCP through the LAMS (endoscopic ultrasound-directed transenteric ERCP) [1]. Endoscopic BD in SAA may also be obtained in cooperation with surgery or interventional radiology (e.g., laparoscopic-assisted ERCP and percutaneous rendezvous) [8,9].

Despite the data in the literature reporting a high percentage of technical and clinical success for endoscopic BD in SAA [10,11], all these kinds of procedures are challenging and not routinely performed by all biliopancreatic endoscopists. For this reason, percutaneous BD, which is a faster and more available procedure, is often preferred to endoscopic BD in patients with SAA [12]. However, it is known that percutaneous BD has long-term drawbacks and cannot be the preferable first-line option.

To our knowledge, there are no current data in Italy about the preferred approach for BD in patients with SAA. The aim of our survey, sent to different centers throughout Italy, was to evaluate the center expertise and the preferred endoscopic BD approach in patients with SAA.

## 2. Materials and Methods

A 34-question survey about the approach to BD in patients with SAA (post-surgical upper-GI reconstructions) was submitted, during a temporal trend of 3 months (April 2023–June 2023), among Italian centers performing interventional biliopancreatic endoscopy (ERCP and interventional EUS), all members of the i-EUS group (available in Appendix A). In 2019, a nationwide educational initiative was held in Italy involving gastroenterologists and GI endoscopists from 40 different centers who were performing interventional EUS. This initiative covered about 80% of the centers that were performing such procedures nationwide at the time. Thus, the i-EUS group (Interventional Endoscopy and Ultrasound) was formed and supported an educational program aimed at improving interventional biliopancreatic procedures and especially optimizing the use of LAMSs in clinical practice [13].

The questionnaire was reviewed by three experts (C.B., A.F. and A.A.) who have significant scientific, clinical and endoscopic experience in the field of biliopancreatic disorders and also in the approach to SAA patients. For some questions, there was the possibility to choose just one of the answers, while, for others, it was allowed to select multiple options. Responses were recorded in an online questionnaire (Google Forms).

All the participants in the present survey explicitly consented to the use of their data for research purposes.

### 2.1. Design of the Questionnaire

The questions were grouped in three different sections:Expertise of endoscopic center: ERCP and EUS quality indicators, availability of additional biliopancreatic services (interventional radiology, surgery) and advanced biliopancreatic endoscopy procedures (device-assisted ERCP, interventional EUS), type of expertise with BD in SAA. Tertiary centers were differentiated from non-tertiary centers according to the number of ERCP procedures performed (fewer or more than 250 per year). Performance measure cut-offs for ERCP and EUS were evaluated according to the European Society of Gastrointestinal Endoscopy (ESGE) quality improvement initiative (minimum standard: bile duct cannulation ≥ 90%, tissue sampling accuracy during EUS-FNA/FNB ≥ 85%, post-ERCP pancreatitis rate < 10%) [14];Biliary drainage approach in the case of Billroth-II reconstruction: type of endoscopic approach (duodenoscope, forward-viewing endoscope, device-assisted ERCP, EUS-guided BD, laparoscopic-assisted ERCP, rendezvous) or percutaneous approach in the case of benign and malignant disease as a first line or as an alternative approach in the case of a previous failed attempt;Biliary drainage approach in the case of Roux-en-Y reconstruction: type of endoscopic approach (duodenoscope, forward-viewing endoscope, device-assisted ERCP, EUS-guided BD, laparoscopic-assisted ERCP, rendezvous) or percutaneous approach in the case of benign and malignant disease as a first line or as an alternative approach in the case of a previous failed attempt.

### 2.2. Statistical Analysis

Categorical variables were summarized with frequencies and proportions. A subanalysis comparing the referral rate and the number of BD procedures performed in patients with SAA between tertiary and non-tertiary centers was performed using the chi-squared test and with the Mann–Whitney test when appropriate (*p* < 0.05 was statistically significant). Statistical analyses were performed using SPSS^®^ 20.0 statistical software (SPSS, Chicago, IL, USA).

## 3. Results

A total of 39 out of 70 Italian centers (response rate: 56%) performing ERCP and interventional EUS completed the survey.

### 3.1. Section I: Expertise of Endoscopic Centers

The majority of participant centers carry out a high number (≥250/year) of ERCP procedures (79%) and have excellent quality indicators for both rate of post-ERCP pancreatitis and biliary cannulation. Considering the number of ERCP procedures per year (<250/year), 8 centers out of 39 were classified as non-tertiary centers. The majority of centers declared the presence in their institution of additional biliopancreatic services such as interventional radiology (92.3%) and biliopancreatic surgery (94.9%). All the participant centers have expertise in ERCP and interventional EUS, especially in the most frequently performed procedures (e.g., drainage of peripancreatic fluid collections, gallbladder drainage and choledochoduodenostomy). Interestingly, only 50% of centers have availability in their endoscopy of device-assisted enteroscopy, suggesting a lower diffusion in Italy of this type of procedure compared to interventional EUS. In Table 1, the characteristics of participant centers are described.

#### Expertise in Surgically Altered Anatomy

Despite the majority of participant centers performing a high number of ERCP procedures, only almost half of them (48.7%) declared themselves to be reference centers for endoscopic BD in SAA. The declared number of BD procedures in SAA performed by participant centers was variable but lower than 10 procedures per year in 64% of them. As expected, there was a significant difference in the total number of endoscopic BD procedures per year between tertiary and non-tertiary centers (16 vs. 4.9, *p* < 0.02).

Considering the rate of referral to another center for endoscopic BD in SAA, 38% of non-tertiary centers declared that they refer directly to another center compared to 9% of tertiary centers (*p* = 0.05). In the case of a previous failure of endoscopic BD in SAA, the rate of referral to more experienced institutions increases, in particular for non-tertiary centers (62% vs. 22%, *p* = 0.02) (Table 2).

### 3.2. Section II: Biliary Drainage Approach in Case of Billroth-II Reconstruction

Among the participant centers, the endoscopic BD approach to patients with Billroth-II reconstruction is usually performed with a standard duodenoscope or with a forward-viewing scope (e.g., operative gastroscope or pediatric colonoscope) in almost equal percentages (49% and 41%, respectively) in both benign and malignant indications. Two centers (5.2%) declared that they use as a first-line approach an advanced procedure such as DAE-ERCP (Figure 1), whereas another two centers out of 39 declared that they use as a first approach an interventional radiology procedure such as percutaneous BD or percutaneous rendezvous.

Interestingly, the participant centers, in the case of previous failed endoscopic BD, declared a very heterogeneous BD approach for both benign and malignant indications. Advanced endoscopic BD procedures such as EUS-BD and DAE-ERCP are used in 20% and 2–5%, respectively, in benign and malignant indications. Cooperative procedures with surgeons (e.g., laparoscopic-assisted ERCP or surgical rendezvous) and with an interventional radiologist (percutaneous rendezvous) were indicated by a significant proportion (5.2% and 15.4%, respectively) of centers as the alternative of choice for a previous failed endoscopic BD procedure. Finally, percutaneous BD was chosen by one fourth (25.2%) of participant centers as the alternative BD procedure of choice in the case of malignant disease.

### 3.3. Section III: Biliary Drainage Approach in Case of Roux-en-Y Reconstruction

In the case of Roux-en-Y reconstruction (Figure 2), the participant centers declared that they more frequently use as a first approach (38–33%) a forward-viewing scope such as a pediatric colonoscope or a short enteroscope independent of the clinical indication. It is interesting that almost the same percentage of centers (around 35%) declared that they use as a first-line approach percutaneous BD also in the case of a benign indication (35%). More advanced procedures such as EUS-BD and DAE-ERCP are used less frequently in the first instance. It is interesting to note that only 35% of the participant centers that declared that they have in their endoscopy the presence of DAE use this type of procedure in the first instance in the case of Roux-en-Y reconstruction.

In the case of a previous failed attempt at endoscopic BD in Roux-en-Y reconstruction, the participant centers have a heterogeneous approach to BD similar to what happens in the case of Billroth-II reconstruction. However, in this situation, EUS-guided BD including endoscopic ultrasound-directed transenteric ERCP (EDEE) is by far the most used alternative approach to endoscopic BD in both benign and malignant indications (34% and 36%, respectively). Only in this kind of situation is the referral to another center taken into account by 17–20% of participant centers. A more complex procedure from the logistical point of view, such as laparoscopic-assisted ERCP, is performed as an alternative approach in only 8–11% of centers.

## 4. Discussion

This is, to our knowledge, the first survey that aimed to evaluate the BD approach in SAA patients and the prevalence of the different techniques used in endoscopic centers. Only one Japanese survey has been published, more than ten years ago, on this topic, and it focused only on one endoscopic technique with DAE [15].

The present survey was sent to all the members of the i-EUS group, which was constituted in Italy in 2019 and supports an educational program aimed at improving interventional biliopancreatic procedures [13]. The members of the i-EUS group are distributed throughout Italy and include tertiary and non-tertiary centers with different levels of endoscopic expertise. We therefore think that the present survey was sent to centers that could reflect the real-world endoscopic approach to biliopancreatic diseases and consequently to BD in SAA patients. The quality indicators for both ERCP and EUS were optimal for almost the totality of the participant centers, suggesting a high reliability of data coming from them.

Considering the characteristics of the participant centers, it is interesting to note that a novel procedure such as interventional EUS is performed by all the centers, whereas DAE is available in only half of them. This aspect shows how DAE has a low diffusion in Italy, even in tertiary centers, and, consequently, how the absence of DAE in certain centers could impact on the preferred BD approach in patients with SAA. It should also be acknowledged that the i-EUS group is mainly focused on educational programs on interventional EUS, and, probably, the ubiquitous diffusion of interventional EUS in this survey is linked to this, overestimating the diffusion of these procedures throughout Italy. At the same time, it is interesting to highlight how an interventional endoscopic procedure introduced in clinical practice not many years ago has, however, gained a capillary diffusion, suggesting the easy availability and reproducibility of the technique. Similarly, it is necessary to acknowledge that the prevalence of DAE use in the Italian endoscopic centers could be underestimated by the characteristics of the participants of our survey, who are mainly specialized in interventional biliopancreatic procedures.

Our survey explored also the presumed number of endoscopic BD procedures in SAA performed per year in each center. The median number of procedures performed in non-tertiary centers was significantly lower than in tertiary ones (5 vs. 16 procedures/year), suggesting that this kind of procedure is extremely uncommon and that the training, especially in non-tertiary centers, could become difficult. The declared median number of procedures performed in a year also in tertiary centers was not impressive. However, this result should be considered with caution because it is not based on real numbers. Moreover, it is necessary to acknowledge that we decided arbitrarily to stratify the center expertise based on the number of ERCP procedures performed per year (more or fewer than 250 per year). Likewise, it is necessary to specify that the current guidelines do not define endoscopic expertise for these types of procedures, or for interventional EUS, or what is the minimum number of procedures required to obtain adequate experience [14].

It was expected, with this low incidence of procedures in SAA patients, for the rate of referral to more experienced centers to be higher in non-tertiary centers. However, our survey showed that only 38% of non-tertiary centers declared that they immediately refer patients requiring BD in SAA to more experienced centers. This percentage increases to 62% only when a first endoscopic attempt at BD performed in the non-tertiary center has failed. It should be acknowledged that our question on the referral rate was general and independent of the type of clinical situation and not based on real patients. It is therefore possible that this low referral rate was based on more simple cases (e.g., BD in Billroth-II patients), which are approached also in non-tertiary centers. However, given the complexity and the rarity of this kind of disease, it is desirable that centers without adequate expertise and facilities for SAA refer directly to expert ones in order to obtain a clinical success and avoid possible complications.

Endoscopic BD in Billroth-II reconstruction is probably the easiest approach among SAA patients because of the short length of the afferent limb, which allows the identification of the papilla in the majority of cases [16]. Given the short length of the limb, there are different endoscopic BD approaches that could be attempted. Our survey shows that, in the majority of centers (91%), the preferred first-line approaches are performed with a standard duodenoscope or a forward-viewing scope (e.g., operative gastroscope or a pediatric colonoscope), which allow standard transpapillary drainage. This result is in line with other previously published papers, where endoscopic BD was performed with a duodenoscope or a forward-viewing scope [12,16].

However, if the identification of the papilla in Billroth-II reconstruction is easy, its successful incannulation is usually suboptimal because of the inverted position of the papilla and the limited maneuverability of the instrument [17]. When cannulation of the papilla fails, an alternative BD method is necessary. Our survey shows how the alternative BD approach in both benign and malignant disease is extremely heterogeneous. As expected, the DAE-ERCP is performed rarely (2–5%) because the longer length and the frontal view of the scope usually do not add any advantage for the papilla approach compared to the other luminal BD techniques. Interestingly, a high percentage of centers declared that they use as alternative approaches percutaneous BD and other invasive cooperative BD approaches such as laparoscopic-assisted ERCP or percutaneous RV for both benign and malignant indications (33% and 39%, respectively). In our survey, the percentages of percutaneous approaches were superior to those of EUS-BD, which, on the contrary, allows internal BD, which, especially for patients with longer life expectancy (i.e., benign indications), is more tolerated than percutaneous BD [18].

Endoscopic BD in the case of Roux-en-Y reconstruction is usually challenging considering the length of the afferent limb usually does not allow papilla or bilio-digestive anastomosis identification with standard endoscopes [19]. However, the length of the afferent limb is extremely variable and usually not known by the endoscopist [20]. This is probably the reason why almost one third of the participant centers declared that they try, as a first attempt, BD with a forward-viewing scope such as pediatric colonoscope or short enteroscope in order to try to reach the papilla. More advanced procedures such as EUS-BD or DAE-ERCP are less used compared to percutaneous BD, which, in one third of centers, is chosen as the first approach also in the case of benign disease. This result shows that, in the real world practice, the percutaneous BD approach is still frequently used, as also demonstrated by the study of Nennstiel et al. [12], despite the high rate of adverse events and long-term management problems [21,22].

It is interesting to note that, in the case of a previous BD failure, the preferred endoscopic BD approach is EUS-BD, used more than twice the number of times that DAE-ERCP is used (34–36% vs. 12–8%). This result is probably related to the low diffusion in the endoscopic centers of DAE but also suggests a preference among endoscopists for EUS-BD procedures [23]. Moreover, EUS-driven procedures such as EDEE allow the papilla to be more easily reached or bilio-jejunal anastomosis in the case of reintervention for complications (e.g., bleeding) or for further treatments (e.g., multistenting or bile duct stone clearance) [10].

It is evident from our survey that, for both types of surgical reconstruction, there is not an endoscopic BD procedure that clearly prevails over another. This is probably related to the lack of available guidelines, of the heterogenicity of the clinical problems, the availability of endoscopic instruments and personal expertise. Considering EUS-guided BD in SAA, it is usually performed, especially for Roux-en-Y reconstruction, with hepaticogastrostomy, antegrade intervention or EDEE [1], which require further expertise compared to other EUS-BD procedures such as gallbladder drainage or choledochoduodenostomy. This technical consideration probably limits the immediate use as a first approach of these kinds of more advanced EUS-guided interventions, as shown from the results of our survey.

Summarizing, our study has different strengths: first, it is the first survey, to our knowledge, in the literature that explores the BD approach in different endoscopic centers for a specific clinical problem such as SAA; second, it gives to us a picture of the furniture and skills of the Italian endoscopic centers; third, it allows us to understand how the low rate of referral to more experienced centers and the use of percutaneous BD are still significant issues.

It is also necessary to acknowledge some limitations of our study: first, the results of this survey, especially about the endoscopic approach for BD, need confirmation in real patients for evaluation of the real application of the endoscopic procedures in clinical practice and the differences among the procedures in terms of efficacy and adverse events; second, it could be questioned whether the sample representativeness reflects accurately the diversity of therapeutic practices because the survey was sent to a group mainly focused on interventional biliopancreatic procedures. However, the members of the i-EUS group include the majority of advanced endoscopic centers in Italy that have a high level of expertise not only in interventional biliopancreatic procedures but also in other endoscopic and luminal examinations.

## 5. Conclusions

This survey shows how BD in SAA still does not have a homogeneous endoscopic approach among centers and that percutaneous BD is taken into account in a significant proportion of patients, also for benign indications, for whom internal BD that minimizes long-term side effects would be desirable. It is also evident from this survey how BD in SAA is not frequently encountered, especially in non-tertiary centers, with possible consequences for clinical success. This aspect should stimulate peripheral centers to make an early referral to more experienced ones in order to avoid delay in clinical success and/or complications. More data from real-world endoscopic experience are needed in order to evaluate the outcomes of the different available endoscopic BD procedures in SAA patients.

## Figures and Tables

**Figure 1 medicina-60-00472-f001:**
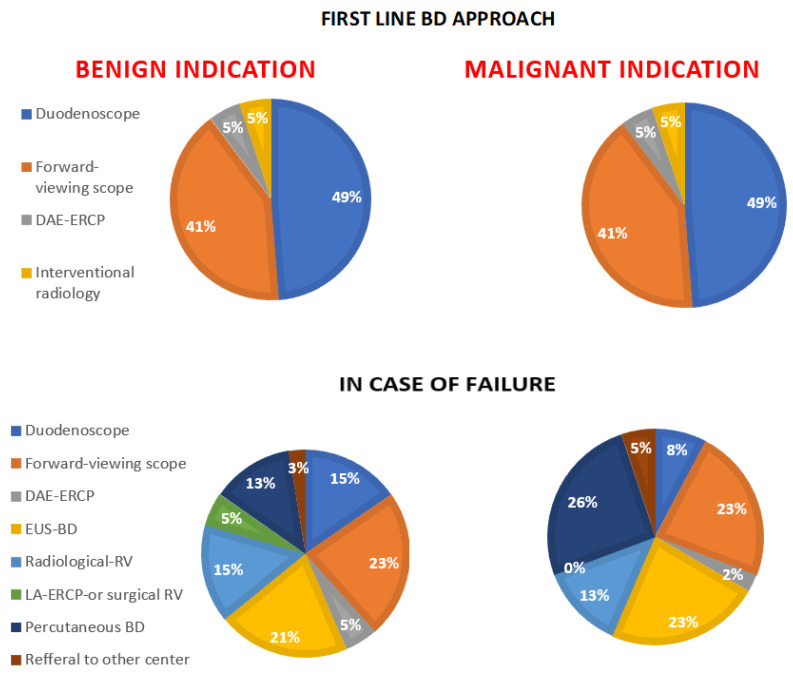
Declared biliary drainage approach in patients with Billroth-II reconstruction as a first-line approach (the two diagrams in the upper side) and in case of failure in both benign (the two diagrams on the left side) and malignant indications (the two diagrams on the right side). BD, biliary drainage; DAE-ERCP, device-assisted enteroscopy–endoscopic retrograde cholangiopancreatography; EUS-BD, endoscopic ultrasound biliary drainage; RV, rendezvous; LA, laparoscopic assisted.

**Figure 2 medicina-60-00472-f002:**
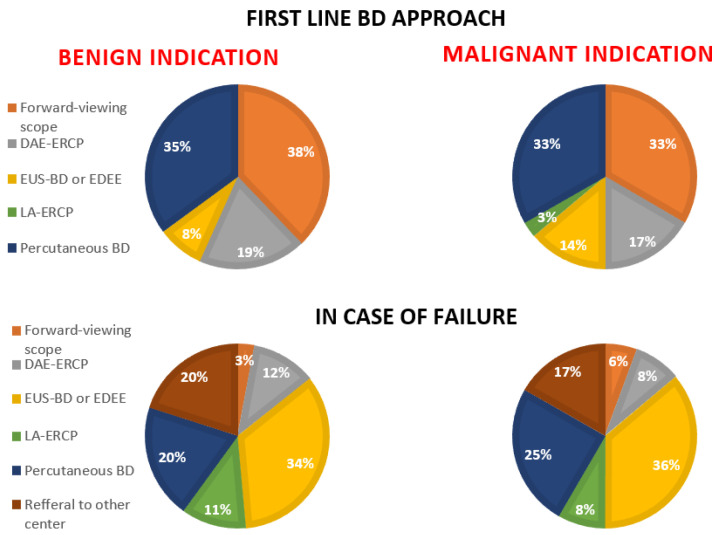
Declared biliary drainage approach in patients with Roux-en-Y reconstruction as a first-line approach (the two diagrams in the upper side) and in case of failure in both benign (the two diagrams in the left side) and malignant indications (the two diagrams in the right side). BD, biliary drainage; DAE-ERCP, device-assisted enteroscopy–endoscopic retrograde cholangiopancreatography; EUS-BD, endoscopic ultrasound biliary drainage; EDEE, endoscopic ultrasound-directed transenteric ERCP; LA, laparoscopic assisted.

**Table 1 medicina-60-00472-t001:** Demographics of the 39 survey participants. Data expressed as percentage or median interquartile range.

Demographics	*n* (%)
Total survey participants	39
ERCP number per year	
<100	2.6%
100–250	17.9%
250–400	53.8%
>400	25.6%
Rate of post-ERCP pancreatitis	
<5%	69.2%
5–10%	30.8%
Biliary cannulation rate	
80–90%	2.6%
>90%	97.4%
EUS-FNA/FNB number per year	
<250	46.2%
250–500	30.8%
500–750	15.4%
>750	7.7%
Rate of FNA/FNB sampling accuracy	
<75%	2.6%
75–85%	7.7%
85–90%	23.1%
>90%	66.7%
Interventional EUS number per year	
<10	17.9%
10–20	20.5%
20–50	23.1%
>50	38.5%
Type of interventional EUS performed	
Peripancreatic collections	100%
Gallbladder drainage	78.9%
Choledochoduodenostomy	81.6%
Gastroenterostomy	50%
Hepaticogastrostomy	31.6%
Interventional radiology availability	
Yes	92.3%
No	7.7%
Device-assisted enteroscopy availability	
Yes	51.3%
No	48.7%
Biliopancreatic surgery availability	
Yes	94.9%
No	5.1%
Declared number of endoscopic BD procedures in SAA/year	
Tertiary centers	16 (5–17.5)
Non-tertiary centers	4.9 (3.7–6.2)

ERCP, endoscopic retrograde cholangiopancreatography; EUS-FNA/FNB, endoscopic ultrasound–fine-needle aspiration/biopsy; BD, biliary drainage; SAA, surgically altered anatomy.

**Table 2 medicina-60-00472-t002:** Expertise in surgically altered anatomy between tertiary and non-tertiary centers. Data expressed as median (IQR) or percentage.

	Tertiary Centers	Non-Tertiary Centers	*p* Value
Endoscopic BD procedures per year	16 (5–17.5)	4.9 (3.7–6.2)	*p* <0.02
Referral rate	9%	38%	*p* = 0.05
Referral rate after previous failure of endoscopic BD	22%	62%	*p* = 0.02

IQR, interquartile range; BD, biliary drainage.

## Data Availability

No new data were created or analyzed in this study. Data sharing is not applicable to this article.

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
