# Peer review of "Italian Survey on Endoscopic Biliary Drainage Approach in Patients with Surgically Altered Anatomy"

_medicina, 2024, doi:10.3390/medicina60030472_

Round 1
Reviewer 1 Report
Comments and Suggestions for Authors
The present Italian survey aimed to assess the prevalence of current biliary drainage (BD) techniques in patients previously treated with upper gastrointestinal surgery. Though the study has uncovered some novel findings, such as the lack of standardisation of BD approaches in Italian hospitals, there are some critical points to be carefully evaluated:
1. While the survey captured responses from both tertiary and non-tertiary endoscopic centres, the sample's representativeness may be questioned. All the included centres are members of a national initiative dedicated to interventional endoscopic ultrasound. This could produce a selection bias, not accurately reflecting the diversity of therapeutic practices of all the centres with endoscopy and surgery units, both those involved in this initiative and those that are not. The authors should acknowledge this point as a limitation of the study.
2. The paper mentions that 39 centres participated in the survey, but the total number of centres contacted and the response rate are not disclosed. Please provide the response rate to exclude a non-response bias.
3. Although the survey outlines the prevailing biliary drainage approaches across different post-surgical scenarios, a critical evaluation of clinical outcomes and patient safety associated with each technique is of paramount importance. Comparative efficacy and safety of each approach are warranted to inform evidence-based clinical decision-making and provide valuable insights into the reported results of this survey.
Additional minor critical points:
- - The questionnaire should be included in the "Supplementary Materials" section;
- - Clarification is needed regarding the expertise of the three reviewers mentioned in line 92;
- - The statement regarding consent for data use (lines 95-96) should be made explicit rather than implicit to ensure transparency;
- - Not all centres listed in Table 1 are experts in interventional EUS, as indicated by the "interventional EUS volume per year" section;
- - The inclusion of EUS-FNA/FNB per year should be justified since it is not directly related to the main topic;
- - It is recommended to provide an additional table comparing results between tertiary and non-tertiary centers, including statistical significance;
- - The low declared median number of endoscopic biliary drainage procedures per year from tertiary centers (16) warrants discussion in the "discussion" section;
- - In line 159, please add the percentage out of the total number of 2 centers for clarity;
- - Clarify that the survey assesses the prevalence of biliary drainage techniques rather than evaluating expertise and outcomes (lines 203-204);
- - Specify quality indicators for ERCP and EUS in the "methods" section, providing specific references;
- - The relevance of stating that interventional EUS was performed by all centers while DAE was available in only half of them should be assessed, particularly if centers were chosen from the i-EUS initiative;
- - In lines 242-244, it is necessary to specify that current guidelines do not define a threshold for minimum number of EUS procedures per year.
A thorough review and correction of multiple scattered English orthographic/grammatical errors throughout the manuscript is strongly recommended. Here some of the detected errors:
- It is unclear whether "data" is intended as singular or plural throughout the text;
- lines 130-131: expertise "in" not "on";
- line 192: "referral" not "refferal"
- line 258: "successful", not "success".
Author Response
We thank the reviewer for his constructive and detailed comments. Please find below our point-by-point responses.
Major comments:
- We thank the reviewer for this comment. Selection bias is crucial in survey-based studies. Our survey was sent to the members of the i-EUS group that is an educational program aimed at improving interventional biliopancreatic procedures. The name of group is probably confusing but it means Interventional Endoscopy AND Ultrasound and therefore is not focused only on EUS. Moreover, we think that this is not causing a selection bias because the members of i-EUS group include the majority of advanced endoscopic centers in Italy that have a high level of expertise not only in interventional EUS but also in other biliopancreatic and luminal procedure. We acknowledged this point in the discussion
- We contacted 70 centers and therefore the response rate was 55% (result added in the text). We think that our response rate was sufficient to have a global estimate of the endoscopic approach to SAA patients throughout Italy.
- We agree with the reviewer that clinical efficacy and safety of any endoscopic biliary drainage procedure in patients with surgical altered anatomy are fundamental. However, this was not the aim of the study because was not based on patients’ data but on a general approach of any endoscopic center. We acknowledged in the discussed that studies, both retrospective or prospective, are needed in order to evaluate the differences in the efficacy and safety between the different types of techniques. We thank the reviewer for this comment because we finished the retrospective multicenter analysis of a very large cohort of SAA patients who underwent to endoscopic BD procedure that we will present at the next ESGE days in Berlin.
Minor comments
- We added our questionnaire in the supplementary material
- We clarified the expertise of the three authors that reviewed the questionnaire
- We modified the statement regarding the consent form
- We agree with you that not all centers are expert in interventional EUS. Indeed, in our survey we included non-tertiary centers that have a low volume of interventional EUS procedures
- We included the EUS-FNB volume because it is a quality indicator of an endoscopic center although not directly related to the main topic of our survey
- We added the suggested table comparing the differences between tertiary and non-tertiary centers
- We comment this point in line 276-283. However, we think that these results should be considered with caution because not based on real patients and specific studies are needed to evaluate the real volume of procedure for each center.
- We specified that the 2 centers were out of 39
- We agree with your comment. We modify accordingly the sentence
- We added the quality indicators for ERCP and EUS accordingly to ESGE guidelines (reference added)
- We agree with your point and we added a comment in lines 269-271. However, it is necessary to specify that the participants of our survey represent the majority of the most experience endoscopic centers in Italy.
- We specified that also current guidelines do not define a threshold for minimum number of EUS procedures per year (lines 301-306).
Quality of English Language
We thank the reviewer for the English revision. We corrected the reported error and we reviewed all the text modifying the grammatical errors.
Reviewer 2 Report
Comments and Suggestions for Authors Dear Author, I read the manuscript "Italian survey on endoscopic biliary drainage approach in patients with surgically altered anatomy" with great interest. The manuscript describes, reporting the data from a questionnaire administered to numerous endoscopic centers, the state of the art in Italy regarding the first-line and alternative BD approach to SAA patients with benign or malignant obstruction. Although the interest in this paper may appear limited to readers from a single geographic area, the most relevant aspect of this paper is the emphasis on the referral of patients to tertiary centers for the treatment of their conditions. The paper was well written, the Material and Methods section and the results section provide all the information necessary for readers to comprehend the author's work adequately. Tables and figures are suitable to resume the author's finding. References are adequate in number. There are no major remarks. Minor remark Discussion- A discussion of strenghts and limitations of your work would help the reader to better understand the importance of the manuscript.
Author Response
We thank the reviewer for his/her kind comment. We added in the discussion a section with strenghts and limitations of the study (lines 341-356)
Reviewer 3 Report
Comments and Suggestions for Authors
patients with surgically altered anatomy (SAA) The authors presented a very interesting topic regarding BD that can be obtained endoscopically with different techniques or with a percutaneous approach. Although the paper is very organised i would suggest to insist on different techniques regarding benign and malign pathologies and explain the diagrames in detail, this would be very useful for practitioners. Also maybe describe some potential complications of each technique and outline which one was the most safe for each type of surgical altered anatomy, furthermore this could be highlighted also differentiate benign from malign diseases and the most efficient technique. I would also suggest replace the diagrams with some graphics. Good luck, a very nive and useful topic.
Comments on the Quality of English LanguageEnglish is ok.
Author Response
We thank the reviewer for her/his appropriate comment. It is very interisting to evaluate the safety and efficacy of the different endoscopic BD procedures especially considering the differences among benign and malignant indications. However, this was not the aim of the study because was not based on patients’ data but on a general approach of any endoscopic center. We acknowledged in the discussed that studies, both retrospective or prospective, are needed in order to evaluate the differences in the efficacy and safety between the different types of techniques. We have finished the retrospective multicenter analysis of a very large cohort of SAA patients who underwent to endoscopic BD procedure that we will present at the next ESGE days in Berlin and that will be part of another study.
We clarified the legend of the figures.
Round 2
Reviewer 1 Report
Comments and Suggestions for Authors
Dear Authors,
thank you for your efforts in improving the present paper. This manuscript is acceptable for publication in its current form.